# The Impact of the COVID-19 Pandemic on Students’ Mental Health and Sleep in Saudi Arabia

**DOI:** 10.3390/ijerph18179344

**Published:** 2021-09-04

**Authors:** Azizah Alyoubi, Elizabeth J. Halstead, Zoe Zambelli, Dagmara Dimitriou

**Affiliations:** 1Sleep Education and Research Laboratory (SERL), Department of Psychology and Human Development, University College London-Institute of Education, 25 Woburn Square, London WC1H 0AA, UK; azizah.alyoubi.17@ucl.ac.uk (A.A.); l.halstead@ucl.ac.uk (E.J.H.); zoe.zambelli.18@ucl.ac.uk (Z.Z.); 2The National Institute for Stress, Anxiety, Depression and Behavioural Change (NISAD), 252 21 Helsingborg, Sweden

**Keywords:** university students, pandemic, depression, stress, anxiety, resilience, sleep

## Abstract

Background: Mental health problems are prevalent among university students in Saudi Arabia. This study aimed to investigate the impact of the COVID-19 pandemic on university students’ mental health and sleep in Saudi Arabia. Method: A total of 582 undergraduate students from Saudi Arabia aged between 18 and 45 years old (M = 20.91, SD = 3.17) completed a cross-sectional online questionnaire measuring depression, anxiety, stress, resilience, and insomnia during the COVID-19 pandemic (2020). Analysis included an independent samples *t*-test, one-way ANOVA, and Hierarchical regression analysis. Results: Undergraduate students reported high levels of depression, anxiety, and perceived stress and low levels of resilience (*p* < 0.001) during the pandemic. In addition, students reported experiencing insomnia. A hierarchical regression analysis indicated that lower resilience, high levels of insomnia, having a pre-existing mental health condition, and learning difficulties (such as dyslexia, dyspraxia, or dyscalculia) were significantly associated with high levels of depression and stress. In addition, lower resilience, a high level of insomnia, and pre-existing mental health conditions were significantly associated with high levels of anxiety. Finally, a lower level of psychological resilience and a high level of insomnia were significantly associated with increased levels of depression, anxiety and stress within university students. Conclusion: This study has provided evidence that a lower level of psychological resilience and insomnia were associated with mental health problems among undergraduate students in Saudi Arabia, thus enhancing psychological resilience and interventions to support sleep and mental health are vital to support student well-being outcomes throughout the pandemic.

## 1. Introduction

In December 2019, Wuhan, a city in China, became the center of an outbreak of the coronavirus disease 2019 (COVID-19) [1], which is caused by severe acute respiratory syndrome coronavirus 2 (SARS-CoV-2) [2]. The World Health Organization (WHO) declared this disease a Public Health Emergency of International Concern (PHEIC) at the end of January 2020 [3]. Recommendations by WHO included taking simple precautions such as physical distancing, wearing a mask, keeping rooms well ventilated, avoiding crowds, and cleaning hands [3]. The first case of COVID-19 was reported in Saudi Arabia in March 2020 [4]. On 8 March 2020, the Saudi Government declared temporary closure of all educational, religious, and social settings and venues (except for essential services such as pharmacies, supermarkets, and hospitals). Further restrictions were imposed on 23 March 2020 that included a curfew starting from 7 pm to 6 am for 21 days and to observe new rules of recommendations by the WHO. Efforts were made to switch teaching among higher education from face to face to online teaching. To aid this, academic staff received training workshops to support students in successful online learning [5]. However, university and school closures have affected young individuals‘ mental health in numerous countries and led to increased levels of anxiety and loneliness [6]. Based on findings from previous studies during pandemics, the WHO recognized that imposing measures such as social distancing might increase anxiety, stress, and anger among individuals [7]. For example, increased levels of anxiety were significantly associated with several avoidance behaviors, such as international travel and visiting public places, during the outbreak in Jeddah, western Saudi Arabia [8].

To date, the impact of COVID-19 on university students’ mental health and sleep has been investigated in several countries including Bangladesh, China, U.S., Australia, and Canada. One study, which collected data using the event-specific distress scale, reported that 69.31% of university students in Bangladesh had a mild to severe level of psychological impact caused by the pandemic [9]. Specifically, home quarantine, the absence of physical exercise, uncertainty about the pandemic trajectory, lack of information, and fear of becoming infected with COVID-19 were determined as risk factors for mental health status among university students in Bangladesh [9]. Additionally, fear of the possibility to be infected and the perceived high risk of becoming infected were also identified to impact university students’ mental health in China [10]. Previous research has shown students who enrolled in the final year at university who had short sleep durations (<6 h per night) were more likely to experience Post-Traumatic Stress Disorder (PTSD) and symptoms of depression among undergraduate students in China [11]. In France, the level of anxiety had increased among university students who did not relocate or move house to a rural area since the beginning of the quarantine period [12]. A study by Goodman, Wang, Guadrrama, and Santana (2020) [13] demonstrated that 60% of students living in the U.S. reported a negative impact of the pandemic on their mental health, while 90% of students were more anxious about finances. Sixty-two percent of students living in the U.S. met the clinical cut-off for depression, while (47%) students met the clinical for generalized anxiety [13]. Additionally, the risk of depression symptoms was three times higher between students who had confirmed and suspected COVID-19 cases among relatives [14]. Australian university students reported 66.3% poorer well-being overall during the pandemic [15].

Age and gender have been identified as risk factors for poorer mental health outcomes during the pandemic. For example, the prevalence of generalized anxiety disorder, symptoms of depression and stress were significantly higher in younger adults than adults over 60 years old in Canada during the SARS-CoV-2 pandemic [16]. In addition, previous studies in numerous countries have revealed that females were at a high of risk having mental health conditions and sleep problems more than males. For instance, female Brazilian students aged between 11 and 19 years old have been reported to have more difficulties falling asleep due to worrying (45.1%) than their male student peers [17,18]. Analysis of focus group data conducted in Germany found that adolescent and young adult females (mean age 22.5 years old) have reported a higher prevalence of depression, and severity of symptoms of depression, than their male peers [19]. Similarly, female graduates (aged between 21 and 40 years old) in Japan reported higher levels of psychological distress, including anxiety, as measured by the General Health Questionnaire-12 (GHQ-12) when compared with male peers during the pandemic [20].

Learning difficulties such as dyslexia, dyspraxia, and dyscalculia have been identified as risk factors for mental health conditions within university students. Prior to the pandemic, Italian university students with learning difficulties such as dyslexia reported high depression scores and lower self-esteem and resilience [21]. Likewise, university students in Georgia aged between 18 and 25 years old who were diagnosed with dyslexia had higher anxiety and depressive symptoms than students who did not have learning difficulties [22].

Several studies have suggested that the prevalence of mental health problems in university students in Saudi Arabia was high prior to the SARS-CoV-2 pandemic. Al Bahhawi et al. (2018) [23] found that 34.3% of university students at Jazan University in Saudi Arabia have experienced stress, 65.7% anxiety, and 53.6% have experienced moderate depression. However, in general, there is limited evidence available on the mental health of university students in Saudi Arabia with only a few studies available that focus on medical students, reporting high levels of stress [24].

Previous literature has reported the association between mental health problems and sleep problems in medical students in Saudi Arabia with risk factors such as low income and academic year identified. Poor sleep quality has been found in 63.2% of students who experienced significant levels of depression, anxiety, and stress (42%, 53%, and 31%, respectively) [25]. In addition, severe social anxiety disorder has been found in first and second academic year medical students associated with a low income [26], and the third academic year group was associated with higher stress and depression among medical students [27].

Due to the high rates of mental health problems reported amongst university students during the pandemic, the need to consider country specific provision and access to mental health services, along with established interventions to treat mental health problems, is justified. For example, a systematic review and meta-analysis examined the effectiveness of psychological interventions for university students experiencing or at risk of developing common mental health problems such as anxiety and depression. The review revealed cognitive and behavioral therapy was the most commonly investigated intervention and was effective for depression and anxiety disorders in this population [28]. Associated factors such as resilience have been demonstrated to improve mental health outcomes in university student population. For example, the Resilience and Coping Intervention (RCI) was conducted on college students aged between 18 and 23 years old and revealed that RCI was an effective resilience intervention for use with college students and reducing the levels of stress and depression [29].

### The Current Study

As services for the treatment of mental health and sleep problems remains limited in Saudi Arabia, this study aims to provide justification for the consideration of improved services and interventions to support both mental health and sleep in this population. To our knowledge, studies have yet to explore mental health and sleep outcomes associated with the impact of the COVID-19 pandemic on Saudi university students beyond medical students. Thus, this study aimed to investigate undergraduate students’ mental health and sleep in Saudi Arabia during the pandemic and determine the factors associated with lower mental health outcomes among university students. We hypothesized that the level of depression, anxiety, and perceived stress would be higher within students demonstrating lower resilience, high levels of insomnia, and who have pre-existing mental health conditions.

## 2. Materials and Methods

### 2.1. Design

This study was a cross-sectional online survey. An online survey was conducted in the Arabic language via Qualtrics, a survey management website (Qualtrics, Provo, UT, USA).

### 2.2. Participants

Participants were eligible if they were currently enrolled as undergraduate students at any university in Saudi Arabia and aged 18 years old or older. Participants who aged under 18 years old and who enrolled in masters and PhD programs were excluded. A total of 582 undergraduate students completed the online questionnaire. All participants self-reported/identified as Arabs from Saudi Arabia aged between 18 and 45 years (M = 20.91, SD = 3.17). See Table 1 for details on the students’ demographic information.

### 2.3. Procedure

Ethical approval was obtained from the University College London, Institute of Education research ethics committee Z6364106/2020/03/105. Participants were recruited via social media channels such as Twitter, Facebook, Instagram, and WhatsApp. Numerous Saudi universities were contacted and provided with an information sheet, advertisement of the study, and the link to the online survey and asked to promote to undergraduate students to participate. The participants first read the information sheet and signed the informed consent and then proceeded to complete the online survey. Participants answered a series of demographic questions and validated measures on mental health (depression; PHQ-9, anxiety; Gad-7, resilience; CD-RISC-25, stress; PSS-10) and sleep (insomnia; ISI). Finally, the participants registered into the prize draw to win a shopping voucher worth £25.

### 2.4. Measures

All questionnaires selected had been previously validated in the Arabic language and used within a Saudi population. All demographics questions and COVID-19-related questions were translated to the Arabic language and back-translated from Arabic to English procedure with cross-cultural considerations. The survey included four sections: (i) demographics and background measures; (ii) COVID-19-related questions related to self-isolating, shielding, whether diagnosed with COVID-19, know anybody who has died from COVID-19, mental health and sleep been impacted since COVID-19, receiving support for a mental health condition before the pandemic, sought support from mental health services since COVID-19; (iii) mental health measures; and (iv) sleep assessment using the Insomnia Severity Index (ISI) measure and questions associated with sleep. These questions were converted into categorical variables.

Do you believe you have a problem with your sleep?Are you currently taking melatonin supplements for your sleep?How would you rate your sleep problem? (very mild, mild, moderate, severe, very severe).

#### 2.4.1. Demographic Information

Participants completed demographic information related to age, gender, ethnicity, education, living area (urban, suburban, rural), occupation status, and medical history. Health behaviors questions enquired about general health, physical health conditions, smoking, and alcohol consumption (See Table 1).

#### 2.4.2. COVID-19 Questions

Participants were asked COVID-19 pandemic related questions. These questions were converted into categorical variables and were used as independent variables. COVID-19 pandemic related questions were as follows:Have you moved house since COVID-19 started?Are you currently self-isolating?Have you been diagnosed with COVID-19?Do you know anybody who has died from COVID-19?Has your mental health been impacted since COVID-19?Has your sleep been impacted in any way?Before COVID-19, were you receiving support for a mental health condition?Since COVID-19, have you sought support from mental health services?

#### 2.4.3. Mental Health Measures

*Depression*: Depression was measured using the patient health questionnaire (PHQ-9) [30]. The questionnaire comprises of nine items to assess depressive symptoms. The PHQ-9 scores can range between 0 to 27 because each item of PHQ- 9 can be scored from 0 (not at all) to 3 (nearly every day). The PHQ-9 rates Minimal depression (0–4), Mild depression (5–9), Moderate depression (10–14), Moderately severe depression (15–19), and Severe depression (20–27). It is designed to assess depression based on the Diagnostic and Statistical Manual of Mental Disorders, Fourth Edition (DSM-IV) criteria (American Psychiatric Association, 2000). The PHQ-9 is a validated and reliable measure within a depressive population for mental health research. The PHQ-9 has been used previously in a university students population in Saudi Arabia and demonstrated good reliability (Cronbach’s alpha was 0.857) [31]. The PHQ-9 also demonstrated excellent internal reliability with a Cronbach’s alpha of 0.847 in this study.

*Anxiety*: Generalized Anxiety Disorder (GAD) scale is designed to assess anxiety symptoms [32]. GAD scale consists of seven items and reflects all Diagnostic and Statistical Manual of Mental Disorders, Fourth Edition (DSM-IV) symptom criteria for GAD. For each item the response option is not all = 0, several days = 1, more than half the days = 2, and nearly every day = 3. The GAD-7 scale score ranges from 0 to 21. The level of Anxiety Severity GAD-7 Scale Score is minimal (0–4), Mild (5–9), Moderate (10–14), and Severe (15–21). The GAD-7 has been used previously in a university student population in Saudi Arabia with Cronbach’s alpha of 0.83 [33]. The GAD-7 showed excellent internal reliability with a Cronbach’s alpha of 0.883 in this study.

*Resilience*: Resilience was measured by Connor-Davidson Resilience Scale [34]. CD-RISC-25 includes 25 items to assesses five components of resilience. The range of the total scale is 0–100 with higher totals indicating more resilience. Responses are shown on a five-point Likert-Type Scale (0 = not at all true to 4 = true nearly all the time). CD-RISC-25 is a validated instrument. The CD-RISC-25 has been used previously in an adult population in Saudi Arabia and Cronbach’s alpha was 0.955 [35]. The CD-RISC-25 demonstrated excellent internal reliability with a Cronbach’s alpha of 0.918 in this study.

*Stress*: Perceived stress scale (PSS-10) [36] consists of 10 items to measures the degree to which one perceives aspects of one’s life as uncontrollable, unpredictable, and overloading. PSS scores range from 0 to 40. Participants are asked to respond to each question on a 5-point Likert scale ranging from 0 = never, 1 = almost never, 2 = sometimes 3 = fairly often to 4 = very often. PSS-10 scores are obtained by reversing responses to the four positively stated items (4,5,7,8). The PSS-10 rated low stress (0–13) moderate stress (14–26) high perceived stress (27–40). The PSS-10 has been used previously in a university student population in Saudi Arabia, however reliability of the measure was unavailable [37]. The PSS-10 demonstrated good internal reliability with a Cronbach’s alpha of 0.678 in this study.

#### 2.4.4. Sleep

*Sleep assessment*: The Insomnia Severity Index (ISI) [38]. The ISI is a self-report instrument measuring perceptions of insomnia. It has been used as both a brief screening instrument and a treatment outcome measure. The ISI has seven items that assess the severity of sleep-onset and sleep maintenance difficulty, satisfaction with sleep, impairments in daily functioning, and degree of distress and concern caused by insomnia. A higher value indicates more severe symptoms of insomnia. The total score of ISI scale was interpreted as normal (0–7), subthreshold (8–14), moderate (15–21), and severe insomnia (22–28). The ISI has been used in an adult population previously in Saudi Arabia, however reliability of the measure was unavailable [39]. The ISI demonstrated good internal reliability with a Cronbach’s alpha of 0.799 in this study.

### 2.5. Data Analysis

Data were analyzed using SPSS statistics version 25. The tests of normality were conducted for all continuous variables using Kolmogorov-Smirnov test and Shapiro-Wilk test which indicated that the data were normally distributed. A descriptive analysis was conducted to evaluate the means and standard deviations of responses received for different questions. Pearson correlation coefficients were used to explore the association between demographic and mental health outcomes. An independent samples *t*-test and One-way ANOVA were employed to determine the differences in mean scores between demographic and COVID-19 variables and mental health outcomes. Hierarchical regression analysis was used for depression, anxiety, and perceived stress models. In these models, age and gender were entered as the first regression step. Physical health conditions, learning difficulties, and pre-existing mental conditions were entered in the second regression step for depression and anxiety models. Physical health conditions, learning difficulties, pre-existing mental conditions, and exercises were entered in the second regression step for perceived stress model. Isolation, sleep, and resilience were entered into the third regression step for depression, anxiety, and perceived stress models. A power calculation determined the minimum number of participants needed in order to detect a small effect size was 85. This was based on conducting a linear regression test with five predictor variables, assuming statistical power of 0.80 and an alpha level of 0.05 [40].

## 3. Results

In the sample, 73% were female and 25% were male. The majority (78%) of undergraduate students were enrolled at the King Abdulaziz University and 16% were studying at Jeddah University in Saudi Arabia. Most of the participants (91%) lived in urban areas. The majority of participants (93%) reported to be living with parents and guardians. Twenty percent of participants had physical health conditions, while 4% had learning difficulties such as dyslexia, dyspraxia, and dyscalculia.

### 3.1. Students’ Mental Health during COVID-19

Pearson correlation coefficient analysis indicated that undergraduate students who reported a significantly high level of depression also reported high level of anxiety, and high level of perceived stress and low level of resilience (*p* < 0.001). Older undergraduate students reported lower depression levels (*r =* −0.109, *p* = 0.008)*,* lower anxiety levels (*r =* −0.158, *p* < 0.001) than younger students. There was a negative and no significant correlation in the level of perceived stress between older and younger students. Older undergraduate students reported more resilience levels (*r* = 0.095, *p* = 0.021) than younger students.

*Gender*: Female students reported significantly higher level of stress (*t* (573) = −3.894, *p* < 0.001), as well as significantly higher level of depression (*t* (573) = −4.669, *p* < 0.001) and higher level of anxiety (*t* (573) = −3.855, *p* < 0.001) than male students. Male students had higher levels of resilience (*t* (573) = 2.638, *p* = 0.009) than female students.

Table 2 presents the results of an independent samples *t*-test to analyze the association between pre-existing mental health conditions and physical health conditions and mental health outcomes. Students with pre-existing mental health conditions reported significantly higher level of anxiety (*t* (537) = 8.936, *p* < 0.001), depression (*t* (537) = 8.990, *p* < 0.001), and perceived stress (*t* (537) = 7.617, *p* < 0.001) than students in the study whose did not have pre-existing mental health conditions. Students with pre-existing mental health conditions reported significantly lower psychological resilience (*t* (537) = −5.013, *p* < 0.001) than students whose did not have pre-existing mental health conditions (see Table 2). Undergraduate students who reported existing physical health conditions had higher level of depression (*t* (568) = 4.124, *p* < 0.001), higher anxiety (*t* (568) = 3.617, *p* < 0.001) higher stress (*t* (568) = 2.625, *p* = 0.009). Nonetheless, there was no significant difference in resilience scores among students who reported physical health condition and those who did not (see Table 2).

### 3.2. COVID-19 Related Variables and Psychological Impact

Since the COVID-19 pandemic, 8% of Saudi students have moved house permanently, while 34% moved house temporarily; 37.1% were self-isolating and 65.8% were shielding; thus, were not leaving home for any reason for at least 12 weeks to reduce the risk of infection. The percentage of students who were diagnosed with COVID-19 was only 1.0%, however 37.3% of students knew someone who died from COVID-19.

Nearly one-third of students reported that their mental health worsened since COVID-19 as they felt more anxiety 22.2%, depression 25.4%, stress 17.9%, and uncertainty 9.5%.

Over one-third of students reported that sleep had been impacted and worsened during the pandemic. For example, 22% having trouble falling asleep, 17.9% waking up during the night, 8.8% waking up early in the morning, 25.9% have poor sleep quality, 22.7% feeling tired during the day, 9.3% have nightmares and 5.2% have poor dreams, and 2.4% have hallucinations. Only 8.9% of students were receiving support for mental health conditions before COVID-19. However, 5.7% of students have sought support from mental health services since COVID-19 and reported it had helped them, and 16.8% had not sought mental health support because they did not think it could help them and 51.5% of students did not seek support from mental health services but they stated a different reason for this.

Table 2 presents the results of an independent samples *t*-test to analyze the association between self-isolating and mental health outcomes. Students who were self-isolating reported significantly higher levels of depression (*t* (492) = 3.528, *p* < 0.001), anxiety (*t* (492) = 3.440, *p* = 0.001), stress (*t* (492) = 2.124, *p* = 0.034) than non-isolated students. However, there were no significant difference in the level of resilience between these two groups. Students who received support from mental health services which ceased and were offered online help and declined before COVID-19 reported a significantly higher level of perceived stress *F* (5, 555) = 3.515, *p* = 0.004). Furthermore, those who were receiving support from mental health services which was stopped during the pandemic reported a significantly higher level of depression *F* (5, 555) = 2.770, *p* = 0.018). Interestingly, students who did not seek support from mental health services because they didn’t think it would help reported significant levels of depression *F* (3, 438) = 18.115, *p* < 0.001), anxiety *F* (3, 438) = 11.191, *p* < 0.001) and less psychological resilience *F* (3, 438) = 8043, *p* < 0.001). Students who sought support from mental health services but they did not help them, were experiencing higher stress level *F* (3, 438) = 5.808, *p* = 0.001).

### 3.3. Sleep Disturbances during COVID-19

This study showed that over half of students reported sleep disruption and 1.4% were taking melatonin supplements for the sleep problem. A small percentage (4.3%) had very mild sleep problem, while 16% reported a mild sleep problem, 21.8% moderate, 9.3% severe, and 1.2% very severe. There was no significant difference between older and younger students on the level of insomnia. Students reported a statistically high level of insomnia with a high level of depression, anxiety, perceived stress (*p* < 0.001). Undergraduate students reported statistically more insomnia symptoms with a lower level of psychological resilience (*r* = −0.150, *p* < 0.001).

Table 2 presents the results of an independent samples *t*-test to analyze the association between pre-existing mental health conditions and physical health conditions and Insomnia. The level of insomnia was slightly higher among female students compared with males, but the difference was not statistically significant. Students with pre-existing mental health conditions reported a significantly higher level of insomnia (*t* (537) = 6.507, *p* < 0.001) than other students. Students who reported physical health conditions had significantly higher levels of insomnia (*t* (568) = 3.759, *p* < 0.001) than students who do not have physical health conditions.

Table 2 presents the results of an independent samples *t*-test to analyze the association between self-isolating and insomnia. The level of insomnia was slightly higher for students who self-isolated but there was no significant difference in the level of insomnia between students.

Students who received support from mental health services but had it stopped and offered online and declined before COVID-19 reported a significantly higher level of insomnia *F* (5, 555) = 2.768, *p* = 0.018). Students who sought support from mental health services, but they did not help them, were experiencing higher insomnia level *F* (3, 438) = 4.091, *p* = 0.007).

### 3.4. Predictors of Mental Health Problems during COVID-19

Hierarchical regression was used to explore predictors of mental health conditions (depression, anxiety, and perceived stress). Specifically, depression, anxiety, perceived stress variables, age, and gender were entered as the first regression block. Physical health conditions, learning difficulties, and pre-existing mental conditions variables were entered into the second regression block for depression, anxiety models. Physical health conditions, learning difficulties, and pre-existing mental conditions and physical activity were entered in the second regression block for perceived stress model. Isolation, sleep, and resilience were entered into the third regression block for depression, anxiety, and perceived stress models.

The results of hierarchical regression analysis to explore predictors of depression, anxiety, and perceived stress are displayed in Table 3. The hierarchical regression analysis showed that psychological resilience, insomnia, and pre-existing mental health conditions had a significant association with mental health conditions (depression, anxiety, and perceived stress). In addition, learning difficulties such as dyslexia, dyspraxia, dyscalculia had a significant association with depression and perceived stress. Final predictors level explained 44.0% of the variance in depression. R^2^ = 0.440 (adjusted R^2^ = 0.432, *F* (8, 573) = 56.303, *p* < 0.001), 36.6% of the variance in anxiety. R^2^ = 0.366 (adjusted R^2^ = 0.357, *F* (8, 573) = 41.398, *p* < 0.001), 23.0% of the variance in stress. R^2^ = 0.230 (adjusted R^2^ = 0.218, *F* (9, 572) = 18.992, *p* < 0.001). Upon examining the individual predictors of depression, Insomnia (β = 0.471, *p* < 0.001), psychological resilience (β *=* −0.250, *p* < 0.001), Learning difficulties (β = −0.140, *p* < 0.001) and having pre-existing mental health conditions (β = −0.117, *p* = 0.001) were significantly associated with the level of depression. Examination of the individual predictors of anxiety showed that insomnia (β = 0.387, *p* < 0.001), psychological resilience (β = −0.244, *p* < 0.001), and having pre-existing mental health conditions (β = −0.189, *p* < 0.001) were significantly associated with the level of anxiety. Furthermore, the individual predictors of perceived stress, indicated that psychological resilience (β = −0.285, *p* < 0.001), Insomnia (β = 0.235, *p* < 0.001). Students with pre-existing mental health conditions (β = −0.117, *p* = 0.004) and learning difficulties (β = −0.084, *p* = 0.027) were significantly associated with level of perceived stress.

The current study did not find being isolated during the COVID-19 to be a significant predictor of the level of depression, anxiety, and perceived stress. Furthermore, physical activity and exercise levels were not predictive variables of level of perceived stress.

## 4. Conclusions

The current study aimed to investigate undergraduate students’ mental health and sleep in Saudi Arabia during the pandemic and determine the factors associated with lower mental health outcomes among university students. The present study found that mental health problems such as high levels of depression, anxiety, and perceived stress were associated with each other in university students during COVID-19. These findings are similar to previous studies conducted during the pandemic in several countries including Bangladesh, China, U.S., Australia, and Canada. Specifically, 69.31% of university students in Bangladesh had a mild to severe level of psychological distress and 62% of students in the U.S. met the clinical cutoff for depression, while 47% of university students met the clinical for generalized anxiety due to the pandemic [9,10,13]. Likewise, college students in China have been reported to have experienced high levels of anxiety due to the COVID-19 outbreak [41].

The results of this study also indicated that the socio-demographic characteristics, specifically age and gender, were significantly associated with mental health and resilience. Younger undergraduate students were found to have a higher level of anxiety, depression, stress, and a lower level of resilience than their male peers. This finding is similar to evidence from a previous study that indicated younger adults in China reported a significantly higher prevalence of depressive symptoms and anxiety than older adults during COVID-19 [42]. In addition, a systematic review also found that adolescents aged between 13 and 17 years old in several nations reported higher rates of depression, anxiety, and stress compared with older adolescents during the pandemic [43].

In this study, female students reported higher scores on anxiety, depression, stress a lower level of resilience than their male peers. These findings are supported by several previous studies that have indicated this gender difference during the pandemic. Young adult females have reported a higher severity of symptoms of depression and higher levels of psychological distress (including anxiety) than their young adult male peers in Germany, Japan, and Saudi Arabia [19,20,44]. In addition, a systematic review also found that the COVID-19 pandemic had negatively impacted females psychological and mental health, more than males [45]. Similarly, female students at the University of Zaragoza in Spain reported more stress due to the pandemic [46].

Our study found that undergraduate students had a high level of insomnia which was associated with an increased levels of depression, anxiety, and stress. This finding is consistent with previous literature conducted both prior to and during the pandemic. Medical students in Saudi Arabia who reported sleep disruption and insomnia, also had a higher level of depression, anxiety, and stress prior to the pandemic [25,47]. Similarly, Spanish college students aged between 18 and 42 years old were also found to have insomnia associated with symptoms of depression, anxiety, and stress during the pandemic [48]. Finally, a study conducted during the pandemic on medical students in China confirmed the association between insomnia, perceived stress, and depression [49]. Thus, cognitive behavioral therapy for insomnia (CBT-I) has been found as an effective therapy in reducing insomnia and improving sleep quality. CBT-I comprises non-pharmacological cognitive, behavioral, and educational strategies. Techniques consist of sleep hygiene, such as keeping the bedroom dark, quiet, and cool; stimulus control, such as waking up at the same time every morning and getting out of bed within 10 to 15 min upon awakening; and sleep restriction [50].

This study also found significant associations between psychological resilience and mental health outcomes specifically depression, anxiety, and stress. Undergraduate students who had a lower level of resilience during the pandemic had higher levels of depression, anxiety, and perceived stress. This finding is consistent with the previous studies conducted on university students prior to and during the pandemic. International university students aged between 18 and 59 years old from Australia, the U.S., and Hong Kong with low levels of resilience reported higher levels of psychological distress [51]. Similarly, a study conducted during the pandemic on Swiss university students aged between 21 and 30 years confirmed that students with low levels of resilience were more likely to have depressive symptoms compared to students with high resilience [52]. A study conducted on college students in China also found a negative association between resilience and depressive symptoms during the pandemic [53].

Pre-existing mental health conditions were also significantly associated with mental health status in this study. Having pre-existing mental health conditions lead to increased depressive symptoms, anxiety, and perceived stress within undergraduate students, suggesting pre-existing mental health conditions were a significant predictor of students’ mental health during the pandemic. This finding was also reported by a study conducted on medical students in the U.S. during the pandemic [54].

Learning difficulties such as dyslexia, dyspraxia, and dyscalculia were also found as significant risk factors associated with mental health outcomes [21,22]. The rate of depression and perceived stress were significantly elevated during the pandemic among students with learning difficulties such as dyslexia, dyspraxia, and dyscalculia. A possible explanation for this result might be that students with pre-existing mental health conditions and students with learning difficulties have limited access to essential interventions and mental health services during the pandemic and lockdown in Saudi Arabia.

However, the current study found that physical health conditions were not significantly associated with the levels of depression, anxiety, and stress. This finding is contradictory to a previous study conducted prior to the pandemic. University students in Bangladesh with a mean age of 21.2 years who reported physical health conditions also had a higher level of depression and anxiety [55]. Furthermore, this study found that physical activity and exercise were not predictive variables of the level of perceived stress. This finding seems to contradict previous studies that indicated less physical exercise was significantly associated with stress, anxiety, and depression among university students in Bangladesh prior to and during the outbreak [9,56]. Finally, this study found no significant association between self-isolation during the pandemic and the level of mental health problems (depression, anxiety, and stress).

### 4.1. Limitations and Strengths

The current study indicated important findings that lower resilience, high levels of insomnia, having a pre-existing mental health condition and learning difficulties (such as dyslexia, dyspraxia, and dyscalculia) were significantly associated with high levels of depression and stress among university students in Saudi Arabia during the pandemic. Furthermore, lower resilience, a high level of insomnia, and pre-existing mental health conditions were significantly associated with high levels of anxiety within students in Saudi Arabia.

However, this study has several limitations. First, all participants were self-reported/identified as Arab ethnicity, thus it is limited in generalizability. Another limitation that undergraduate students from King Abdulaziz University and Jeddah University in Saudi Arabia provided data in this study which may not be representative of all student population in Saudi Arabia. Furthermore, it is not possible to generalize the findings to all universities students as the focus of this study was on undergraduate students only. Thus, the evidence that mental health conditions and sleep problems are prevalent among university students during the Covid-19 pandemic emphasizes the importance of future research on this topic. Future research is needed to investigate the long-term impacts of the pandemic on students’ mental health involving numerous education levels and several regions in Saudi Arabia. An important limitation to consider is that there was no data collected pre-pandemic for comparison to look at specific changes related to the pandemic in this population, therefore not all findings can be directly related to the pandemic. Finally, this research utilized self-report measures to assess mental health and sleep problem. However, self-report measures were validated (Arabic version) and widely used in the students’ population.

### 4.2. Implications

The findings of this study may be of significance to enhance our understanding of how the COVID-19 pandemic affects students’ mental health and sleep. Overall, these findings support previous studies conducted with undergraduate students that depression, anxiety, and perceived stress are associated with each other during the COVID-19 outbreak. Specifically, the findings of this study indicated the association that undergraduate students who had a lower level of psychological resilience during the pandemic also had higher levels of depression, anxiety, and perceived stress. The current study also indicated that insomnia is associated with a higher level of mental health conditions, which supporting findings both prior to and during the pandemic. Certain interventions may be beneficial in supporting well-being outcomes in university students, for example, resilience-building interventions are available online, and these could be adapted and evaluated to enhance resilience in university students in Saudi Arabia to support mental health outcomes [57]. In addition, effective treatment for insomnia such as Cognitive Behavioral Therapy for insomnia may be beneficial in alleviating sleep problems [58]. Although these interventions may be beneficial, seeking and accessing mental health services and the provision of interventions for university students remains limited in Saudi Arabia. This study has suggested that it is important to monitor university students’ mental health during the outbreak. In keeping with previous research, psychiatrists and mental health professionals are essential to mitigate the risk of developing mental health problems during the pandemic [59,60], and it is important university students are aware of support and resources available to them within universities and more widely. Our results suggest university students with pre-existing mental health problems are at risk and along with the general population should be considered high priority for support during the pandemic [61]. Country-specific mental health systems that are faced with challenges during the emergency situations such as the current pandemic and require support and provisions to provide better services as per recommendations by the World Health Organization [62,63].

## Figures and Tables

**Table 1 ijerph-18-09344-t001:** Undergraduate students’ demographic information.

Variable		*n*	Percent
Gender	Male	149	25%
	Female	426	73%
University Students are currently enrolled in	King Abdulaziz university	456	78%
	Jeddah university	94	16%
	Other universities at KSA	28	5%
Year at University	First Year	128	22%
	Second year	212	36%
	Third year	102	18%
	Fourth year	75	13%
	Fifth year	31	5%
	Sixth year	9	1%
Employment Status	Student	513	88%
	Employed (full time)	22	4%
	Employed (part time)	18	3%
	Self employed	7	1%
	Unemployed looking for work	17	3%
	Unemployed not looking for work	4	0.7%
Average Annual Family Household Income	Less than £20,000	219	37%
	£20,000–£29,999	69	12%
	£30,000–£39,999	37	6%
	£40,000–£59,999	22	4%
	£60,000–£79,999	29	5%
	£80,000–£99,999	23	4%
	£100,000–£149,999	20	3%
	More than £149,999	37	6%
Area Students live in	Urban	532	91%
	Suburban	30	5%
	Rural	15	2%
Living Situation	Living with parents/guardians	541	93%
	Living alone	21	3.6%
	Living with students	3	0.5%
	On-campus (Residence Hall	6	1%
	Living with non-student roommates	1	0.2%
Smoking	Students smoke	87	15%
Physical Health Conditions	Students have a psychical conditions	115	20%
Learning Difficulties	Students have Learning difficulties	24	4%

**Table 2 ijerph-18-09344-t002:** An independent samples *t*-test comparing means for mental health outcomes and sleep.

	Pre-Existing Mental Health Conditions				Physical Health Conditions				Self-Isolating			
	Yes	No			Yes	No			Yes	No		
Variables	(M/SD)	(M/SD)	*t*	*p*	(M/SD)	(M/SD)	*t*	*p* Value	(M/SD)	(M/SD)	*t*	*p*
Insomnia	13.55 (5.916)	8.93 (5.589)	6.507	**<0.001**	11.53 (6.152)	9.25 (5.708)	3.759	**<0.001**	10.11 (6.287)	9.55 (5.697)	1.044	0.297
Stress	23.11 (7.822)	16.75 (6.428)	7.617	**<0.001**	19.29 (7.778)	17.41 (6.614)	2.625	**0.009**	18.53 (7.041)	17.18 (6.931)	2.124	**0.034**
Resilience	55.32 (20.242)	66.35 (17.028)	−5.013	**<0.001**	62.20 (18.601)	65.41 (17.448)	−1.737	0.083	64.77 (17.888)	65.41 (17.491)	−0.400	0.689
Depression	15.30 (5.889)	8.83 (5.689)	8.990	**<0.001**	12.06 (6.182	9.43 (6.092)	4.124	**<0.001**	11.19 (6.513)	9.21 (5.929)	3.528	**<0.001**
Anxiety	13.08 (5.38)	7.12 (5.283)	8.936	**<0.001**	9.90 (6.193)	7.76 (5.550)	3.617	**<0.001**	9.19 (5.952)	7.43 (5.405)	3.440	**0.001**

Note. Significant (*p* < 0.05) associations between variables are in bold.

**Table 3 ijerph-18-09344-t003:** Hierarchical regression predicting mental health conditions.

	Depression					Anxiety					Perceived Stress				
Variables	*B*	*SE*	*β*	*R^2^*	*p*	*B*	*SE*	*β*	*R^2^*	*p*	*B*	*SE*	*β*	*R^2^*	*p*
Block 1				0.035					0.040					0.017	
Age	−0.189	0.080	−0.097		0.018	−0.267	0.074	−0.148		0.000	−0.110	0.090	0.050		0.223
Gender	1.988	0.533	0.153		0.000	1.469	0.493	0.122		0.003	1.699	0.600	0.117		0.005
Block 2				0.147					0.150					0.087	
Age	−0.145	0.075	−0.075		0.054	−0.227	0.070	−0.125		0.001	−0.084	0.087	−0.39		0.334
Gender	1.609	0.506	0.124		0.002	1.071	0.468	0.089		0.023	1.410	0.584	0.097		0.016
Physical health conditions	−0.852	0.522	−0.065		0.103	−0.770	0.483	−0.063		0.112	−0.193	0.604	−0.13		0.749
Learning difficulties	−3.280	0.909	−0.143		0.000	−0.934	0.842	−0.044		0.268	2.404	1.050	0.094		0.022
Pre-existing mental health conditions	−2.637	0.431	−0.251		0.000	−2.925	0.399	−0.300		0.000	2.389	0.498	0.204		0.000
Block 3				0.440					0.366					0.230	
Age	−0.075	0.062	−0.039		0.222	−0.166	0.061	−0.092		0.007	−0.013	0.081	0.006		0.870
Gender	0.998	0.412	0.077		0.016	0.584	0.407	0.048		0.152	0.926	0.540	0.064		0.087
Physical health conditions	−0.140	0.426	−0.011		0.742	−0.216	0.421	−0.018		0.607	0.275	0.559	0.019		0.623
Learning difficulties	−3.224	0.740	−0.140		**<0.001**	−0.869	0.730	−0.041		0.235	2.156	0.969	0.084		**0.027**
Pre-existing mental health conditions	−1.231	0.360	−0.117		**0.001**	−1.839	0.356	−0.189		**<0.001**	1.370	0.472	0.117		**0.004**
Self-isolating	−0.464	0.275	−0.053		0.092	−0.160	0.271	−0.020		0.555	−0.313	0.360	0.032		0.385
Resilience	−0.088	0.011	−0.250		**<0.001**	−0.079	0.011	−0.244		**<0.001**	−0.112	0.015	0.285		**<0.001**
Insomnia	0.498	0.034	0.471		**<0.001**	0.379	0.034	0.387		**<0.001**	0.277	0.045	0.235		**<0.001**
Exercise vs. no Exercise	-	-	-		-	-	-	-		-	0.889	0.499	0.066		0.075

Note. Significant (*p* < 0.05) associations between variables are in boldface.

## Data Availability

The dataset supporting reported results is available by request to the contact author.

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
