# Peer review of "The Impact of the COVID-19 Pandemic on Students’ Mental Health and Sleep in Saudi Arabia"

_ijerph, 2021, doi:10.3390/ijerph18179344_

Round 1
Reviewer 1 Report
Thank you for the chance to read this interesting and engaging paper. Overall I found the data engaging and persuasive, it was an honor to be allowed to support the authors in improving this research paper.
There are several issues that, when addressed, will improve the paper under review.
Most critical issues:
1. The authors present recommendations for practice, interventions, etc. in the conclusion (implications section) that must be integrated into the introductory sections. The justifications for recommending specific interventions is currently not supported by the data being presented, as no data regarding interventions was collected/presented.
2. The introduction includes a long paragraph on mental health among medical students which confuses the issue since no medical students are included in the present sample. I recommend deleting this material and focusing on the literature specific to the sample being discussed here. A sentence or two regarding the fact that medical students are better represented in the literature is sufficient to convey the basic idea. Clarifying the introduction will make the substantive sections more clear.
3. Minor grammar issues throughout (subject-verb agreement, missing words, definiteness) need to be cleaned up. A list of these issues are included below.
4. I find issues with the use of tables in the article. First, my preference would be for an explicit description of each table prior to the table in the text. Following the form: "Table 2 presents..." Second, Table 1 presents demographics with some demographics being reported, but having very small numbers of responses compared to the N, either the reported characteristics need to be explained in more depth, or questions with low response rates need to be removed. Knowing that 14 of the 582 respondents either drink alcohol or prefer not to say adds nothing to the novel and interesting results being reported. Third, tables 2 and 3 are so tightly organized that it makes these tables extremely difficult to read. I suggest either a) breaking up tables to better present specific information, or b) including tables in landscape formatting, or making other formatting changes that improve legibility.
5. Paragraph 2 in the Discussion section includes an overly broad description of the findings, including both age, gender, and issues related to sleep. The findings concerning age and gender need to be better situated in the literature, and would benefit from being separated from the findings on sleep/insomnia.
6. The inclusion of information about a systematic review and meta-analysis is included in the Implications section. This needs to be deleted or moved to the introduction, no new information should be included here. Any recommendations for practice need to be integrated into the 'story' that the data indicates. I suggest removing all recommendations for practice that are not explicitly and clearly linked to the data collected and the measures under discussion. I do not see any reported material concerning what kinds of interventions work with this specific population, which makes these recommendations particularly unwelcome.
List of grammar issues to correct:
Page 2, line 63: "Several studies has..." to "Several studies have..."
Page 2, line 67-68: "A high level of stress was found" to "A high level of stress has been found..."
Page 2, line 69-70: Missing word, "...with a great majority [of what?] being conducted with the general population..."
Table 1's title: "students'" needs to be capitalized. Table 2's and Table 3's titles needs to be capitalized appropriately, use title case for all table names.
Page 6, line 228: "Table2" needs a space between Table and the number.
Page 6, line 240-241: Please move the parenthetical to report after each item in the ordered list and correct "uncertain" to "uncertainty": "...more anxiety [insert (22.5%)], depression [insert appropriate %], stress [insert appropriate %], and uncertain [replace with uncertainty] about different situations."
Page 7, line 270 (but also check the full text for other examples): The sample is exclusively made up of undergraduate students, there is no need to continue to specify that fact. Please delete "undergraduate" in most cases and keep "students" or "respondents."
Page 10, line 380: Change "affecting" to "affects"
Author Response
Thank you for the chance to read this interesting and engaging paper. Overall I found the data engaging and persuasive, it was an honour to be allowed to support the authors in improving this research paper.
There are several issues that, when addressed, will improve the paper under review.
Most critical issues:
- The authors present recommendations for practice, interventions, etc. in the conclusion (implications section) that must be integrated into the introductory sections. The justifications for recommending specific interventions is currently not supported by the data being presented, as no data regarding interventions was collected/presented.
Thank you for this comment, we have revised the discussion section to align with the findings and integrated intervention material into the introduction:
Introduction text revision:
" Due to the high rates of mental health problems reported amongst university students during the pandemic, the need to consider country specific provision and access to mental health services, along with established interventions to treat mental health problems, is justified. For example, a systematic review and meta-analysis examined the effectiveness of psychological interventions for university students experiencing or at risk of developing common mental health problems such as anxiety and depression. The review revealed cognitive and behavioural therapy was the most commonly investigated intervention and was effective for depression and anxiety disorders in this population [27]. Associated factors such as resilience have been demonstrated to improve mental health outcomes in university student population. For example, the Resilience and Coping Intervention (RCI) was conducted on college students aged between 18 and 23 years old and revealed that RCI was an effective resilience intervention for use with college students and reducing the levels of stress and depression [28]."
Implication section revision in discussion:
“The findings of this study may be of significance to enhance our understanding of how the COVID-19 pandemic affects students’ mental health and sleep. Overall, these findings support previous studies conducted with undergraduate students that depression, anxiety and perceived stress are associated with each other during the COVID-19 outbreak. Specifically, the findings of this study indicated the association that undergraduate students who had a lower level of psychological resilience during the pandemic also had higher levels of depression, anxiety, and perceived stress. The current study also indicated that insomnia is associated with a higher level of mental health conditions, which supporting findings both prior to and during the pandemic. Certain interventions may be beneficial in supporting well-being outcomes in university students, for example, resilience-building interventions are available online and these could be adapted and evaluated to enhance resilience in university students in Saudi Arabia to support mental health outcomes [55]. In addition, effective treatment for insomnia such as Cognitive Behavioural Therapy for Insomnia may be beneficial in alleviating sleep problems [56]. Although these interventions may be beneficial, seeking and accessing mental health services and the provision of interventions for university students remains limited in Saudi Arabia.”
The introduction includes a long paragraph on mental health among medical students which confuses the issue since no medical students are included in the present sample. I recommend deleting this material and focusing on the literature specific to the sample being discussed here. A sentence or two regarding the fact that medical students are better represented in the literature is sufficient to convey the basic idea. Clarifying the introduction will make the substantive sections more clear.
Thank you for this comment, we have amended this paragraph to the following:
" Previous literature has reported the association between mental health problems and sleep problems in medical students in Saudi Arabia with risk factors such as low income and academic year identified. Poor sleep quality has been found in 63.2% of students who experienced significant levels of depression, anxiety, and stress (42%, 53%, and 31% respectively) [25]. In addition, severe social anxiety disorder has been found in first and second academic year medical students associated with a low income [26] and the third academic year group was associated with higher stress and depression among medical students [25]."
Thank you, we have corrected the grammar issues through proofreading and those identified below.
4. I find issues with the use of tables in the article. First, my preference would be for an explicit description of each table prior to the table in the text. Following the form: "Table 2 presents..." Second, Table 1 presents demographics with some demographics being reported, but having very small numbers of responses compared to the N, either the reported characteristics need to be explained in more depth, or questions with low response rates need to be removed. Knowing that 14 of the 582 respondents either drink alcohol or prefer not to say adds nothing to the novel and interesting results being reported. Third, tables 2 and 3 are so tightly organized that it makes these tables extremely difficult to read. I suggest either a) breaking up tables to better present specific information, or b) including tables in landscape formatting, or making other formatting changes that improve legibility.
Thank you for your comments,
First, we have added sentences that describe each table prior to the table reference in the text.
" Table 2 presents the results of an independent samples t-test to analyze the association between pre-existing mental health conditions and physical health conditions and mental health outcomes."
"Table 2 presents the results of an independent samples t-test to analyze the association between self-isolating and mental health outcomes."
"Table 2 presents the results of an independent samples t-test to analyze the association between pre-existing mental health conditions and physical health conditions and Insomnia."
"Table 2 presents the results of an independent samples t-test to analyze the association between self-isolating and Insomnia."
"The results of hierarchical regression analysis to explore predictors of depression, anxiety, and perceived stress are displayed in Table 3."
Secondly, we have deleted questions with low response rates such as prefer not to say In Table 1.
Thirdly, Thank you for your suggestion, we have improved the format of Table 2 and Table 3 in landscape formatting.
5. Paragraph 2 in the Discussion section includes an overly broad description of the findings, including both age, gender, and issues related to sleep. The findings concerning age and gender need to be better situated in the literature, and would benefit from being separated from the findings on sleep/insomnia.
Thank you for this comment, we have updated the discussion section and have separated the findings as suggested.
" The results of this study also indicated that the socio-demographic characteristics, specifically age and gender, were significantly associated with mental health and resilience. Younger undergraduate students were found to have a higher level of anxiety, depression, stress, and a lower level of resilience than their male peers. This finding is similar to evidence from a previous study that indicated younger adults in China reported a significantly higher prevalence of depressive symptoms and anxiety than older adults during COVID-19 [41]. In addition, a systematic review also found that adolescents aged between 13 and 17 years old in several nations reported higher rates of depression, anxiety, and stress compared with older adolescents during the pandemic [42]."
"In this study, female respondents reported higher scores on anxiety, depression, stress a lower level of resilience than their male peers. These findings are supported by several previous studies that have indicated this gender difference during the pandemic. Young adult females have reported a higher severity of symptoms of depression, higher levels of psychological distress (including anxiety) than their young adult males peers in Germany, Japan and Saudi Arabia[19,20,43]. In addition, a systematic review also found that the COVID-19 pandemic had negatively impacted females psychological and mental health, more than males [44]. Similarly, females students at university of Zaragoza in Spain reported more stress due to the pandemic [45]."
" Our study found that undergraduate students had a high level of insomnia which was associated with an increased levels of depression, anxiety, and stress. This finding is consistent with previous literature conducted both prior to and during the pandemic. Medical students in Saudi Arabia who reported sleep disruption and insomnia, also had a higher level of depression, anxiety, and stress prior to the pandemic [25,46]. Similarly, Spanish college students aged between 18 and 42 years old were also found to have insomnia associated with symptoms of depression, anxiety, and stress during the pandemic [47]. Finally, a study conducted during the pandemic on medical students in China confirmed the association between insomnia, perceived stress, and depression [48]."
- The inclusion of information about a systematic review and meta-analysis is included in the Implications section. This needs to be deleted or moved to the introduction, no new information should be included here. Any recommendations for practice need to be integrated into the 'story' that the data indicates. I suggest removing all recommendations for practice that are not explicitly and clearly linked to the data collected and the measures under discussion. I do not see any reported material concerning what kinds of interventions work with this specific population, which makes these recommendations particularly unwelcome.
Thank you for this comment, we have moved the systematic review and meta-analysis in the implication section into the introduction section and revised the implication section as below.
Introduction text revision:
" Due to the high rates of mental health problems reported amongst university students during the pandemic, the need to consider country specific provision and access to mental health services, along with established interventions to treat mental health problems, is justified. For example, a systematic review and meta-analysis examined the effectiveness of psychological interventions for university students experiencing or at risk of developing common mental health problems such as anxiety and depression. The review revealed cognitive and behavioural therapy was the most commonly investigated intervention and was effective for depression and anxiety disorders in this population [27]. Associated factors such as resilience have been demonstrated to improve mental health outcomes in university student population. For example, the Resilience and Coping Intervention (RCI) was conducted on college students aged between 18 and 23 years old and revealed that RCI was an effective resilience intervention for use with college students and reducing the levels of stress and depression [28]."
Implication section revision:
" The findings of this study may be of significance to enhance our understanding of how the COVID-19 pandemic affects students’ mental health and sleep. Overall, these findings support previous studies conducted with undergraduate students that depression, anxiety and perceived stress are associated with each other during the COVID-19 outbreak. Specifically, the findings of this study indicated the association that undergraduate students who had a lower level of psychological resilience during the pandemic also had higher levels of depression, anxiety, and perceived stress. The current study also indicated that insomnia is associated with a higher level of mental health conditions, which supporting findings both prior to and during the pandemic. Certain interventions may be beneficial in supporting well-being outcomes in university students, for example, resilience-building interventions are available online and these could be adapted and evaluated to enhance resilience in university students in Saudi Arabia to support mental health outcomes [55]. In addition, effective treatment for insomnia such as Cognitive Behavioural Therapy for Insomnia may be beneficial in alleviating sleep problems [56]. Although these interventions may be beneficial, seeking and accessing mental health services and the provision of interventions for university students remains limited in Saudi Arabia. "
List of grammar issues to correct:
Page 2, line 63: "Several studies has..." to "Several studies have..."
Page 2, line 67-68: "A high level of stress was found" to "A high level of stress has been found..."
Page 2, line 69-70: Missing word, "...with a great majority [of what?] being conducted with the general population..."
Table 1's title: "students'" needs to be capitalized. Table 2's and Table 3's titles needs to be capitalized appropriately, use title case for all table names.
Page 6, line 228: "Table2" needs a space between Table and the number.
Page 6, line 240-241: Please move the parenthetical to report after each item in the ordered list and correct "uncertain" to "uncertainty": "...more anxiety [insert (22.5%)], depression [insert appropriate %], stress [insert appropriate %], and uncertain [replace with uncertainty] about different situations."
Page 7, line 270 (but also check the full text for other examples): The sample is exclusively made up of undergraduate students, there is no need to continue to specify that fact. Please delete "undergraduate" in most cases and keep "students" or "respondents."
Page 10, line 380: Change "affecting" to "affects"
Thank you for this comment, we have corrected all the grammar issues as listed above and revised the manuscript after proof reading.
Reviewer 2 Report
The paper " University students’ mental health and sleep in Saudi Arabia during COVID-19” is very well written, and the authors deserve great merit.
Here are some suggestions to improve article. Please edit the article based on them.
Title
It seems better to write the toile "The impact of the COVID-19 pandemic on students' mental health and sleep in Saudi Arabia"
Introduction
Line 29:
Beforehand, write a paragraph about coronavirus, where and how it spreads, and WHO recommendations on coronavirus.
Studies related to Covid 19 in different individuals should be added to the literature. The following references can be reported.
https://www.mdpi.com/2071-1050/12/18/7792
https://www.mdpi.com/1660-4601/18/9/4563
https://www.mdpi.com/2227-9067/8/6/438
https://www.mdpi.com/1660-4601/18/12/6304
Line 91:
The main hypothesis of the researcher should be stated here and the expected result should be mentioned
Materials and Methods
Design:
Were the questionnaires compatible with Saudi Arabia in terms of sociology and social culture? What steps were taken to understand this issue? And if this compatibility did not exist, how did the questionnaires adapt?
Participants:
If possible, all Saudi Arabia undergraduate students in this age group, as well as the ratio of men and women in them, should be mentioned as a statistical population.
Please calculate the sample size
Discussion:
Congratulations to the authors on the discussion, it is very well written.
Author Response
The paper " University students’ mental health and sleep in Saudi Arabia during COVID-19” is very well written, and the authors deserve great merit.
Here are some suggestions to improve article. Please edit the article based on them.
Title
It seems better to write the toile "The Impact of the COVID-19 Pandemic on Students' Mental Health and Sleep in Saudi Arabia"
Thank you for this comment, we have changed the title to the below.
"The Impact of the COVID-19 Pandemic on Students' Mental Health and Sleep in Saudi Arabia"
Introduction
Line 29:
Beforehand, write a paragraph about coronavirus, where and how it spreads, and WHO recommendations on coronavirus.
Thank you for this comment, we have added the following sentences about coronavirus into the introduction section.
" In December 2019, Wuhan, a city in China, became the center of an outbreak of the coronavirus disease 2019 (COVID-19) [1], which is caused by severe acute respiratory syndrome coronavirus 2 (SARS-CoV-2)[2]."
" The World Health Organisation (WHO) declared this disease a Public Health Emergency of International Concern (PHEIC) at the end of January 2020 [3]. Recommendations by WHO included taking simple precautions such as physical distancing, wearing a mask, keeping rooms well ventilated, avoiding crowds, and cleaning hands [3]. "
Studies related to Covid 19 in different individuals should be added to the literature. The following references can be reported.
https://www.mdpi.com/2071-1050/12/18/7792
Analysis of Self-Concept in Adolescents before and during COVID-19 Lockdown: Differences by Gender and Sports Activity.
Thank you for this suggestion, our study focused on the impact of COVID-19 on university students mental health and therefore this paper is not directly relevant to our population.
https://www.mdpi.com/1660-4601/18/9/4563
Effect of COVID-19 on Health-Related Quality of Life in Adolescents and Children: A Systematic Review.
Thank you for this suggestion, we have added this reference into the discussion section:
" In addition, a systematic review also found that the COVID-19 pandemic had negatively impacted females psychological and mental health, more than males [44]."
https://www.mdpi.com/2227-9067/8/6/438
Different Effects of the COVID-19 Pandemic on Exercise Indexes and Mood States Based on Sport Types, Exercise Dependency and Individual Characteristics
Thank you for this suggestion, our study focused on the impact of COVID-19 on university students mental health and therefore this paper is not directly relevant to our population.
https://www.mdpi.com/1660-4601/18/12/6304
Potential Improvement in Rehabilitation Quality of 2019 Novel Coronavirus by Isometric Training System; Is There “Muscle-Lung Cross-Talk”?
Thank you for this suggestion, we have added this reference into the introduction section.
"which is caused by severe acute respiratory syndrome coronavirus 2 (SARS-CoV-2)"
Line 91:
The main hypothesis of the researcher should be stated here and the expected result should be mentioned
Thank you for this comment, we have added the following hypothesis.
" As services for the treatment of mental health and sleep problems remains limited in Saudi Arabia, this study aims to provide justification for the consideration of improved services and interventions to support both mental health and sleep in this population. To our knowledge, studies have yet to explore mental health and sleep outcomes associated with the impact of the COVID-19 pandemic on Saudi university students beyond medical students. Thus, this study aimed to investigate undergraduate students’ mental health and sleep in Saudi Arabia during the pandemic and determine the factors associated with lower mental health outcomes among university students. We hypothesized that the level of depression, anxiety and perceived stress would be higher within students demonstrating lower resilience, high levels of insomnia, and who have pre-existing mental health conditions."
Materials and Methods
Design:
Were the questionnaires compatible with Saudi Arabia in terms of sociology and social culture? What steps were taken to understand this issue? And if this compatibility did not exist, how did the questionnaires adapt?
Thank you for this comment, this study is cross-sectional and our online survey was conducted in the Arabic language. We have translated the demographics questions and COVID-19 related questions to the Arabic language. In terms of the measures, mental health measures and sleep assessment had been validated in Arabic language and used in previous studies within Saudi populations.
We have added the following sentences into the materials and methods section.
"All questionnaires selected had been previously validated in the Arabic language and used within a Saudi population. "
"All demographics questions and COVID-19 related questions were translated to the Arabic language and back-translated from Arabic to English procedure with cross-cultural considerations was conducted ."
" The PHQ-9 has been used previously in a university students population in Saudi Arabia and demonstrated good reliability (Cronbach’s alpha was 0.857) [30]."
"The GAD-7 has been used in university students population in Saudi Arabia with Cronbach’s alpha of 0.83 [32]."
" The CD-RISC-25 has been used previously in an adult population in Saudi Arabia and Cronbach’s alpha was 0.955 [34]."
" The PSS-10 has been used previously in a university student population in Saudi Arabia, however reliability of the measure was unavailable [36]."
" The ISI has been used in an adult population previously in Saudi Arabia, however reliability of the measure was unavailable [38]."
Participants:
If possible, all Saudi Arabia undergraduate students in this age group, as well as the ratio of men and women in them, should be mentioned as a statistical population.
Thank you for this comment, we could not find this data and therefore have not been able to include this.
Please calculate the sample size
Thank you for this comment, we have added a power calculation.
" A power calculation determined the minimum number of participants needed in order to detect a small effect size was 85. This was based on conducting a linear regression test with five predictor variables, assuming statistical power of 0.80 and an alpha level of 0.05. "
Discussion:
Congratulations to the authors on the discussion, it is very well written.
Thank you for this comment.
Reviewer 3 Report
This manuscript entitled "University students’ mental health and sleep in Saudi Arabia during COVID-19" aimed to investigate the impact of the COVID-19 pandemic on university students‘ mental health and sleep.
The manuscript is very interesting. However, some issues should be addressed by the authors:
ABSTRACT
- Method section is quite short.
- Please, include more information about the data analysis. How was the interview analyzed? Include this information.
- Result section: include more numerical and statistical information
INTRODUCTION
- It is important for the background to include recent references about mental health prevalence from just before the pandemic and other results from mental health during the pandemic. Please, bring results from other countries in this section.
METHODS
- All intruments/questionnaires were valdiated to Saudi Arabia?
RESULTS
- Table 2 is quite difficult to understand. Please, improve format. Try in a landscape format, I think it could improve.
- Table 3 is even worse to understand.
DISCUSSION
- This section is very short. Please, improve it and discuss more in-depth the results.
- The strong points from the study should be included. I suggest in the same section as Limitation: "Limitation and strond points'
- Conclusion in the last paragraph is not clear and should be rewritten.
REFERENCES
- Several recent articles from IJERPH could be cited.
- Below I suggest some articles pre and during COVID pandemic about mental health in university students' which should improve the introduction and discussion sections:
Escobar, D.F.S.S.; Noll, P.R.S.; Jesus, T.F.; Noll, M. Assessing the Mental Health of Brazilian Students Involved in Risky Behaviors. Int. J. Environ. Res. Public Health 2020, 17, 3647.
Chaabane, S.; Doraiswamy, S.; Chaabna, K.; Mamtani, R.; Cheema, S. The Impact of COVID-19 School Closure on Child and Adolescent Health: A Rapid Systematic Review. Children 2021, 8, 415. https://doi.org/10.3390/children8050415
Seven, Ü.S.; Stoll, M.; Dubbert, D.; Kohls, C.; Werner, P.; Kalbe, E. Perception, Attitudes, and Experiences Regarding Mental Health Problems and Web Based Mental Health Information Amongst Young People with and without Migration Background in Germany. A Qualitative Study. Int. J. Environ. Res. Public Health 2021, 18, 81.
Tahara, M.; Mashizume, Y.; Takahashi, K. Coping Mechanisms: Exploring Strategies Utilized by Japanese Healthcare Workers to Reduce Stress and Improve Mental Health during the COVID-19 Pandemic. Int. J. Environ. Res. Public Health 2021, 18, 131.
Escobar, D.F.S.S.; Jesus, T.F.; Noll, P.R.S.; Noll, M. Family and School Context: Effects on the Mental Health of Brazilian Students. Int. J. Environ. Res. Public Health 2020, 17, 6042.
Jones, E.A.K.; Mitra, A.K.; Bhuiyan, A.R. Impact of COVID-19 on Mental Health in Adolescents: A Systematic Review. Int. J. Environ. Res. Public Health 2021, 18, 2470. https://doi.org/10.3390/ijerph18052470
Author Response
ABSTRACT
- Method section is quite short.
- Please, include more information about the data analysis. How was the interview analyzed? Include this information.
- Result section: include more numerical and statistical information
Thank you for this comment, Abstract has been updated.
"Abstract: Background: Mental health problems are prevalent among university students in Saudi Arabia. This study aimed to investigate the impact of the COVID-19 pandemic on university students‘ mental health and sleep in Saudi Arabia. Method: A total of 582 undergraduate students from Saudi Arabia aged between 18 and 45 years old (M = 20.91, SD = 3.17) completed a cross-sectional online questionnaire measuring depression, anxiety, stress, resilience, and insomnia during the COVID-19 pandemic (2020). Analysis included an independent samples t-test, one-way ANOVA and Hierarchical regression analysis. Results: Undergraduate students reported high levels of depression, anxiety, and perceived stress and low levels of resilience (p <.001) during the pandemic. In addition, students reported experiencing insomnia. A hierarchical regression analysis indicated that lower resilience, high levels of insomnia, having a pre-existing mental health conditions and learning difficulties (such as dyslexia, dyspraxia, dyscalculia) were significantly associated with high levels of depression and stress. In addition, lower resilience, a high level of insomnia, and pre-existing mental health conditions were significantly associated with high levels of anxiety. Finally, a lower level of psychological resilience and a high level of insomnia were significantly associated with increased levels of depression, anxiety and stress within university students. Conclusion: This study has provided evidence that a lower level of psychological resilience and insomnia were associated with mental health problems among undergraduate students in Saudi Arabia, thus enhancing psychological resilience and interventions to support sleep and mental health are vital to support student well-being outcomes throughout the pandemic."
INTRODUCTION
- It is important for the background to include recent references about mental health prevalence from just before the pandemic and other results from mental health during the pandemic. Please, bring results from other countries in this section.
"Thank you for this comment, we have added recent references about mental health prevalence before and during the pandemic into the introduction.
" However, university and school closures has affected young individuals‘ mental health in numerous countries and lead to increased the level of anxiety and loneliness."
" Australian university students reported 66.3% poorer well-being overall during the pandemic. "
" Age and gender have been identified as risk factors for poorer mental health outcomes during the pandemic. For example, the prevalence of generalized anxiety disorder, symptoms of depression and stress were significantly higher in younger adults than adults over 60 years old in Canada during the SARS-CoV-2 pandemic[16]. In addition, previous studies in numerous countries have revealed that females were at a high of risk having mental health conditions and sleep problems more than males. For instance, female Brazilian students aged between 11 and 19 years old have been reported to have more difficulties falling asleep due to worrying (45.1%) than their male student peers [17,18]. Analysis of focus group data conducted in Germany found that adolescent and young adult females (mean age 22.5 years old) have reported a higher prevalence of depression, and severity of symptoms of depression, than their male peers [19]. Similarly, female graduates (aged between 21 and 40 years old) in Japan reported higher levels of psychological distress, including anxiety, as measured by the General Health Questionnaire-12 (GHQ-12) when compared with male peers during the pandemic [20]."
Learning difficulties (such as dyslexia, dyspraxia, dyscalculia) have been identified as risk factors for mental health conditions within university students. Prior to the pandemic, Italian university students with learning difficulties such as dyslexia reported high depression scores and lower self-esteem and resilience [21]. Likewise, university students in Georgia aged between 18 and 25 years old and diagnosed with dyslexia had higher anxiety and depressive symptoms than students whoes did not have learning difficulties [22]."
" Previous literature has reported the association between mental health problems and sleep problems in medical students in Saudi Arabia with risk factors such as low income and academic year identified. Poor sleep quality has been found in 63.2% of students who experienced significant levels of depression, anxiety, and stress (42%, 53%, and 31% respectively) [25]. In addition, severe social anxiety disorder has been found in first and second academic year medical students associated with a low income [26] and the third academic year group was associated with higher stress and depression among medical students [25]. "
METHODS
- All instruments/questionnaires were valdiated to Saudi Arabia?
Thank you for this comment, this study is cross-sectional and our online survey was conducted in the Arabic language. We have translated the demographics questions and COVID-19 related questions to the Arabic language. In terms of the instruments, mental health measures and sleep assessment had been validated in Arabic language and appropriate to use in Saudi population.
We have added the following sentences into the materials and methods section.
"All questionnaires selected had been previously validated in the Arabic language and used within a Saudi population. "
"All demographics questions and COVID-19 related questions were translated to the Arabic language and back-translated from Arabic to English procedure with cross-cultural considerations was conducted ."
" The PHQ-9 has been used previously in a university students population in Saudi Arabia and demonstrated good reliability (Cronbach’s alpha was 0.857) [30]."
"The GAD-7 has been used in university students population in Saudi Arabia with Cronbach’s alpha of 0.83 [32]."
" The CD-RISC-25 has been used previously in an adult population in Saudi Arabia and Cronbach’s alpha was 0.955 [34]."
" The PSS-10 has been used previously in a university student population in Saudi Arabia, however reliability of the measure was unavailable [36]."
" The ISI has been used in an adult population previously in Saudi Arabia, however reliability of the measure was unavailable [38]."
RESULTS
- Table 2 is quite difficult to understand. Please, improve format. Try in a landscape format, I think it could improve.
Thank you for this comment, we have improved the format of Table 2 in landscape formatting.
- Table 3 is even worse to understand.
Thank you for this comment, we have improved the format of Table 3 in landscape formatting.
DISCUSSION
- This section is very short. Please, improve it and discuss more in-depth the results.
Thank you for this comment, the discussion section has been updated.
Discussion section:
"The current study aimed to investigate undergraduate students’ mental health and sleep in Saudi Arabia during the pandemic and determine the factors associated with lower mental health outcomes among university students. The present study found that mental health problems such as high levels of depression, anxiety, and perceived stress were associated with each other in university students during COVID-19. These findings are similar to previous studies conducted during the pandemic in several countries including Bangladesh, China, U.S., Australia and Canada. Specifically, 69.31% of university students in Bangladesh had a mild to severe level of psychological distress and 62% of students in the U.S. met the clinical cutoff for depression, while 47% of university students met the clinical for generalized anxiety due to the pandemic [9,10,13]. Likewise, college students in China have been reported to have experienced high levels of anxiety due to the COVID-19 outbreak [40]."
"The results of this study also indicated that the socio-demographic characteristics, specifically age and gender, were significantly associated with mental health and resilience. Younger undergraduate students were found to have a higher level of anxiety, depression, stress, and a lower level of resilience than their male peers. This finding is similar to evidence from a previous study that indicated younger adults in China reported a significantly higher prevalence of depressive symptoms and anxiety than older adults during COVID-19 [41]. In addition, a systematic review also found that adolescents aged between 13 and 17 years old in several nations reported higher rates of depression, anxiety, and stress compared with older adolescents during the pandemic [42]. "
"In this study, female respondents reported higher scores on anxiety, depression, stress a lower level of resilience than their male peers. These findings are supported by several previous studies that have indicated this gender difference during the pandemic. Young adult females have reported a higher severity of symptoms of depression, higher levels of psychological distress (including anxiety) than their young adult males peers in Germany, Japan and Saudi Arabia[19,20,43]. In addition, a systematic review also found that the COVID-19 pandemic had negatively impacted females psychological and mental health, more than males [44]. Similarly, females students at university of Zaragoza in Spain reported more stress due to the pandemic [45]."
"Our study found that undergraduate students had a high level of insomnia which was associated with an increased levels of depression, anxiety, and stress. This finding is consistent with previous literature conducted both prior to and during the pandemic. Medical students in Saudi Arabia who reported sleep disruption and insomnia, also had a higher level of depression, anxiety, and stress prior to the pandemic [25,46]. Similarly, Spanish college students aged between 18 and 42 years old were also found to have insomnia associated with symptoms of depression, anxiety, and stress during the pandemic [47]. Finally, a study conducted during the pandemic on medical students in China confirmed the association between insomnia, perceived stress, and depression [48]."
"This study also found significant associations between psychological resilience and mental health outcomes specifically depression, anxiety and stress. Undergraduate students who had a lower level of resilience during the pandemic had higher levels of depression, anxiety, perceived stress. This finding is consistent with the previous studies conducted on university students prior to and during the pandemic. International university students aged between 18 and 59 years old from Australia, the U.S., and Hong Kong with low levels of resilience reported higher levels of psychological distress[49]. Similarly, a study conducted during the pandemic on Swiss university students aged between 21 and 30 years confirmed that students with low levels of resilience were more likely to have depressive symptoms compared to students with high resilience [50]. A study conducted on college students in China also found a negative association between resilience and depressive symptoms during the pandemic[51]."
"Pre-existing mental health conditions were also significantly associated with mental health status in this study. Having pre-existing mental health conditions lead to increased depressive symptoms, anxiety and perceived stress within undergraduate students, suggesting pre-existing mental health conditions were a significant predictor of students’ mental health during the pandemic. This finding was also reported by a study conducted on medical students in the U.S. during the pandemic [52]. "
"Learning difficulties such as dyslexia, dyspraxia, dyscalculia were also found as significant risk factors associated with mental health outcomes [21,22]. The rate of depression and perceived stress were significantly elevated during the pandemic among students with learning difficulties such as dyslexia, dyspraxia, dyscalculia. A possible explanation for this result might be students with pre-existing mental health conditions and students with learning difficulties have limited access to essential interventions and mental health services during the pandemic and lockdown in Saudi Arabia."
"However, the current study found that physical health conditions were not significantly associated with the levels of depression, anxiety and stress. This finding is contradictory to a previous study conducted prior to the pandemic. University students in Bangladesh with a mean age of 21.2 years who reported physical health conditions also had a higher level of depression and anxiety [53]. Also, this study found that physical activity and exercise were not predictive variables of the level of perceived stress. This finding seems to contradict previous studies that indicated less physical exercise was significantly associated with stress, anxiety, depression among university students in Bangladesh prior to and during the outbreak [9,54]. Finally, this study found no significant association between self-isolation during the pandemic and the level of mental health problems (depression, anxiety, stress)."
- The strong points from the study should be included. I suggest in the same section as Limitation: "Limitation and strong points'
Thank you for this comment, we have added the following strong points into the limitations.
Limitations and strengths :
" The current study indicated important findings that lower resilience, high levels of insomnia, having a pre-existing mental health conditions and learning difficulties (such as dyslexia, dyspraxia, dyscalculia) were significantly associated with high levels of depression and stress among university students in Saudi Arabia during the pandemic. Also, lower resilience, a high level of insomnia, and pre-existing mental health conditions were significantly associated with high levels of anxiety within students in Saudi Arabia."
"However, this study has several limitations. Firstly, all participants were self-reported/identified as Arab ethnicity, thus it is limited in generalizability. Another limitation that undergraduate students from King Abdulaziz University and Jeddah University in Saudi Arabia provided data in this study which means it is not possible to relate the findings to the general population in Saudi Arabia which has different regions and different universities. Also, it is not possible to generalize the findings to all universities students as the focus of this study was on undergraduate students only. Thus, the evidence that mental health conditions and sleep problems are prevalent among university students during the Covid-19 pandemic emphasizes the importance of future research on this topic. Future research is needed to investigate the long-term impacts of the pandemic on students’ mental health involving numerous education levels and several regions in Saudi Arabia. An important limitation to consider is that there was no data collected pre-pandemic for comparison to look at specific changes related to the pandemic in this population, therefore not all findings can be directly related to the pandemic. Finally, this research utilized self-report measures to assess mental health and sleep problem. However, self-report measures were validated (Arabic version) and widely used in the students' population."
- Conclusion in the last paragraph is not clear and should be rewritten.
Thank you for this comment, we have revised the conclusion in the last paragraph to:
"The findings of this study may be of significance to enhance our understanding of how the COVID-19 pandemic affects students’ mental health and sleep. Overall, these findings support previous studies conducted with undergraduate students that depression, anxiety and perceived stress are associated with each other during the COVID-19 outbreak. Specifically, the findings of this study indicated the association that undergraduate students who had a lower level of psychological resilience during the pandemic also had higher levels of depression, anxiety, and perceived stress. The current study also indicated that insomnia is associated with a higher level of mental health conditions, which supporting findings both prior to and during the pandemic. Certain interventions may be beneficial in supporting well-being outcomes in university students, for example, resilience-building interventions are available online and these could be adapted and evaluated to enhance resilience in university students in Saudi Arabia to support mental health outcomes [55]. In addition, effective treatment for insomnia such as Cognitive Behavioural Therapy for Insomnia may be beneficial in alleviating sleep problems [56]. Although these interventions may be beneficial, seeking and accessing mental health services and the provision of interventions for university students remains limited in Saudi Arabia."
REFERENCES
- Several recent articles from IJERPH could be cited.
- Below I suggest some articles pre and during COVID pandemic about mental health in university students' which should improve the introduction and discussion sections:
Escobar, D.F.S.S.; Noll, P.R.S.; Jesus, T.F.; Noll, M. Assessing the Mental Health of Brazilian Students Involved in Risky Behaviors. Int. J. Environ. Res. Public Health 2020, 17, 3647.
Thank you for your suggestion, we have added this reference into the paper.
" In addition, previous studies in numerous countries have revealed that females were at a high of risk having mental health conditions and sleep problems more than males. For instance, female Brazilian students aged between 11 and 19 years old have been reported to have more difficulties falling asleep due to worrying (45.1%) than their male student peers."
Chaabane, S.; Doraiswamy, S.; Chaabna, K.; Mamtani, R.; Cheema, S. The Impact of COVID-19 School Closure on Child and Adolescent Health: A Rapid Systematic Review. Children 2021, 8, 415. https://doi.org/10.3390/children8050415
Thank you for your suggestion, we have added this reference into the paper.
" However, university and school closures has affected young individuals‘ mental health in numerous countries and lead to increased the level of anxiety and loneliness."
Seven, Ü.S.; Stoll, M.; Dubbert, D.; Kohls, C.; Werner, P.; Kalbe, E. Perception, Attitudes, and Experiences Regarding Mental Health Problems and Web Based Mental Health Information Amongst Young People with and without Migration Background in Germany. A Qualitative Study. Int. J. Environ. Res. Public Health 2021, 18, 81.
Thank you for your suggestion, we have added this reference into the paper.
" Analysis of focus group data conducted in Germany found that adolescent and young adult females (mean age 22.5 years old) have reported a higher prevalence of depression, and severity of symptoms of depression, than their male peers."
Tahara, M.; Mashizume, Y.; Takahashi, K. Coping Mechanisms: Exploring Strategies Utilized by Japanese Healthcare Workers to Reduce Stress and Improve Mental Health during the COVID-19 Pandemic. Int. J. Environ. Res. Public Health 2021, 18, 131.
Thank you for your suggestion, we have added this reference into the paper.
" Similarly, female graduates (aged between 21 and 40 years old) in Japan reported higher levels of psychological distress, including anxiety, as measured by the General Health Questionnaire-12 (GHQ-12) when compared with male peers during the pandemic."
Escobar, D.F.S.S.; Jesus, T.F.; Noll, P.R.S.; Noll, M. Family and School Context: Effects on the Mental Health of Brazilian Students. Int. J. Environ. Res. Public Health 2020, 17, 6042.
Thank you for your suggestion, we have added this reference into the paper.
" For instance, female Brazilian students aged between 11 and 19 years old have been reported to have more difficulties falling asleep due to worrying (45.1%) than their male student peers."
Jones, E.A.K.; Mitra, A.K.; Bhuiyan, A.R. Impact of COVID-19 on Mental Health in Adolescents: A Systematic Review. Int. J. Environ. Res. Public Health 2021, 18, 2470. https://doi.org/10.3390/ijerph18052470
Thank you for your suggestion, we have added this reference into the paper.
" In addition, a systematic review also found that adolescents aged between 13 and 17 years old in several nations reported higher rates of depression, anxiety, and stress compared with older adolescents during the pandemic."
Thank you for your suggestion, We have added further references from IJERPH:
- Liu, C.; McCabe, M.; Dawson, A.; Cyrzon, C.; Shankar, S.; Gerges, N.; Kellett-Renzella, S.; Chye, Y.; Cornish, K. Identifying Predictors of University Students’ Wellbeing during the COVID-19 Pandemic—A Data-Driven Approach. International Journal of Environmental Research and Public Health 2021, 18, 6730.
" Australian university students reported 66.3% poorer well-being overall during the pandemic "
- Nwachukwu, I.; Nkire, N.; Shalaby, R.; Hrabok, M.; Vuong, W.; Gusnowski, A.; Surood, S.; Urichuk, L.; Greenshaw, A.J.; Agyapong, V.I.O. COVID-19 Pandemic: Age-Related Differences in Measures of Stress, Anxiety and Depression in Canada. International Journal of Environmental Research and Public Health 2020, 17, 6366.
"Age and gender have been identified as risk factors for poorer mental health outcomes during the pandemic. For example, the prevalence of generalized anxiety disorder, symptoms of depression and stress were significantly higher in younger adults than adults over 60 years old in Canada during the SARS-CoV-2 pandemic".
Round 2
Reviewer 3 Report
Thanks for your efforts! All my suggestions were addressed.
Congratulation!
Author Response
Dear Reviewer,
Please find the responses to your comments attached. With Kind Regards
